# Molecular and Cellular Responses of DNA Methylation and Thioredoxin System to Heat Stress in Meat-Type Chickens

**DOI:** 10.3390/ani11071957

**Published:** 2021-06-30

**Authors:** Walid S. Habashy, Marie C. Milfort, Romdhane Rekaya, Samuel E. Aggrey

**Affiliations:** 1NutriGenomics Laboratory, Department of Poultry Science, University of Georgia, Athens, GA 30602, USA; milfort@uga.edu; 2Department of Animal and Poultry Production, Damanhour University, Damanhour 22511, Egypt; 3Department of Animal and Dairy Science, University of Georgia, Athens, GA 30602, USA; rrekaya@uga.edu

**Keywords:** heat stress, antioxidant, oxidative damage, thioredoxin, peroxiredoxin

## Abstract

**Simple Summary:**

Performance traits and mortality are negatively affected by heat stress. The responses of chickens to HS are extremely complex. Understanding of the molecular mechanism of DNA methylation and the TXN system under HS may provide a new strategy to mitigate the effect of HS. Based on our results, the thioredoxin pathway system under HS may provide clues to nutritional strategies to mitigate the effect of HS in meat-type chicken.

**Abstract:**

Heat stress (HS) causes molecular dysfunction that adversely affects chicken performance and increases mortality. The responses of chickens to HS are extremely complex. Thus, the aim of this study was to evaluate the influence of acute and chronic exposure to HS on the expression of thioredoxin–peroxiredoxin system genes and DNA methylation in chickens. Chickens at 14 d of age were divided into two groups and reared under either constant normal temperature (25 °C) or high temperature (35 °C) in individual cages for 12 days. Five birds per group at one and 12 days post-HS were euthanized and livers were sampled for gene expression. The liver and *Pectoralis major* muscle were sampled for cellular analysis. mRNA expression of thioredoxin and peroxiredoxins (Prdx) 1, 3, and 4 in the liver were down-regulated at 12 days post-HS compared to controls. The liver activity of thioredoxin reductase *(TXNRD)* and levels of peroxiredoxin1 *(Prdx1)* at 12 days post-HS were significantly decreased. The results reveal that there was a significant decrease in DNA methylation at 12 days post HS in liver tissues. In conclusion, pathway of thioredoxin system under HS may provide clues to nutritional strategies to mitigate the effect of HS in meat-type chicken.

## 1. Introduction

Heat waves and global climate change affect poultry negatively, so the understanding of how birds respond to high temperatures is important to the development of strategies to alleviate the lowered productivity and animal well-being. Heat stress (HS) impacts feed consumption, growth performance, feed conversion ratio, and mortality of broiler chickens [1,2,3]. We distinguish two categories of HS, the first being acute HS which refers to short-term rise in ambient temperature and the second being chronic HS which refers to a sustained long-term rise in ambient temperature. Acute HS is more likely under production environments, as the ambient temperatures tend to cool down during the nighttime whereas chronic HS can last anywhere from an entire day to a few weeks.

HS has been shown to be one of the causal factors of oxidative stress in broilers and in white leghorns [4,5,6]. Oxidative stress is caused by the overproduction of reactive oxygen species (ROS) due to disturbed homeostasis between pro-oxidant and antioxidant systems where the antioxidants are depressed compared to pro-oxidants [7,8]. ROS induce oxidative damage to biomolecules such as lipids, proteins, and nucleic acids resulting in lipid peroxidation, protein carbonyls, and oxidized guanine (8-oxo-dG) among others [9]. Oxidative stress also results in stable or heritable changes to gene expression that do not affect the DNA sequence, referred to as epigenetic changes, by altering DNA methylation [10,11]. The changes in DNA methylation induced by oxidative stress are mainly due to effects on the DNA methyltransferase enzymes that directly methylate DNA leading to both global hypomethylation and site-specific hypermethylation [12]. Our current knowledge about the level of DNA methylation in birds under HS remains scant.

Due to the fact that oxidative stress can seriously damage animal cells, even at the epigenetic level, there are multiple cellular defense systems to counteract or neutralize undesired ROS. Thioredoxin (TXN) and peroxiredoxin (Prdx) proteins are part of such systems. The Thioredoxin-Peroxiredoxin (TXN–Prdx) system scavenges hydrogen peroxides generated through mitochondrial cellular respiration and superoxide dismutase enzyme activity [13,14]. Prdx reduces H_2_O_2_ to water, becoming oxidized in the process on Cys 47, which then forms an intermolecular disulfide with Cys 170 [15]. Oxidized Prdx is then reduced back to its active form by TXN, which acts as an immediate electron donor [15]. Of the six peroxiredoxins identified in vertebrates, two (and Prdx5) are missing in chickens [16]. The ubiquitous TXN system is constituted of the proteins TXN and thioredoxin reductase (TR, TXNRD) and the cofactor NADPH [17]. TXN interacts with ROS damaged proteins such as oxidized Prdx through a pair of redox-active cysteines, thereby restoring their enzymatic activity by reducing oxidized active site cysteines [18]. The oxidized TXN is then reduced back to its active form as an electron donor by TXNRD, with NADPH being a required co-factor in the reaction [19]. The effects of elevated ambient temperatures on the TXN system genes are not clear. Understanding of the molecular mechanism of DNA methylation and the TXN system under HS may provide a new strategy to mitigate the effect of HS. Therefore, the objective of this study was to evaluate the influence of short- and long-term exposure to HS on DNA methylation and the gene expression and cellular activity of TXN–Prdx system in chicken liver and *Pectoralis* (*P*.) *major* muscle.

## 2. Materials and Methods

### 2.1. Experimental Design and Animals

The research herein was conducted under guidelines approved by the University of Georgia’s institutional animal care and use committee. A total of 48 male Cobb500 (fast growing broilers) chickens were divided into two groups and raised in constant environments of either normal or high temperature (25 °C or 35 °C) from days 14 to 26 of age in individual cages and fed a diet comprising 18.7% CP and 3560 Kcal ME/K ad libitum. At 1-day post HS (day 15) and 12 days post HS (day 26), tissues samples from the *P. major* muscle and liver were collected from 5 birds of each treatment and were directly placed in liquid nitrogen and stored later at −86 °C.

### 2.2. Determination of DNA Methylation (ng/mL)

We extracted DNA from tissue samples of P. major and liver tissues with Trizol reagent (Invitrogen Corp., Carlsbad, CA, USA) following manufacturer’s instructions and stored the extract at −20 °C until further processing. DNA samples were then converted to single strands by heating to 95 °C for 5 min followed by rapid chilling on ice. At that point, the DNA was digested with 5 units of nuclease P1 (Sigma, Saint Louis, MO, USA, item No. 8630). The pH was adjusted to 7.5–8.5 using 1 M Tris and 5 µL of alkaline phosphatase was then added and followed by incubation for 45 min at 37 °C and boiling for 10 min. The lysate was then stored at 4 °C until determination of methylation. The concentration of DNA methylation, measured with a commercial kit (Cayman Chemical Company, Ann Arbor, MI, USA, Item No. 589324), was determined spectrophotometrically using a Spectra Max 5 microplate reader (Molecular Devices, Sunnyvale, CA, USA) at 405 nm.

### 2.3. Gene Expression of the Thioredoxin System

For gene expression analysis, liver samples were collected from 5 birds from each treatment at 1 and 12 days post HS and were immediately placed in liquid nitrogen and later stored at −86 ˚C. Total RNA was extracted from liver tissues using Trizol reagent (Invitrogen Corp., Carlsbad, CA, USA) according to the manufacturer’s instructions and purified with RNeasy mini kits (Qiagen, Valencia, CA, USA) according to the manufacturer’s protocol. The RNA samples were suspended in RNase-free water and sample purity and concentration were measured on a NanoDrop spectrophotometer (Thermo Scientifc, Wilmington, DE, USA). For cDNA synthesis, 10 μg of total RNA was reverse transcribed with high capacity cDNA reverse transcription kits according to manufacturer’s protocol (Applied Biosystems, Carlsbad, CA, USA). The genes studied were: *thioredoxin* (*TXN*), *thioredoxin reductase* (*TXNRD1*), and *peroxiredoxin-1* (*PRDX1*), *peroxiredoxin-3* (*PRDX3*), and *peroxiredoxin-4* (*PRDX4*).

Real-time PCR reactions were performed using the StepOnePlus (Applied Biosystems, Carlsbad, CA, USA). Final concentration of 47.1 ng/µL cDNA served as a template in a 20 µL PCR mixture containing a final concentration of 150 nM from 10 µM primer stocks and Fast SYBR Green Master Mix (Applied Biosystems, Carlsbad, CA, USA). The PCR conditions were 95 °C for 20 s, followed by 40 cycles of 95 °C for 3 s and 60 °C for 30 s. In addition, at the end of each reaction, a melting temperature curve of every PCR reaction was determined. Data were analyzed according to the 2^−ΔΔCt^ method [20] and were normalized by β-actin expression in each sample. The National Center for Biotechnology Information (NCBI) accession numbers, forward, reverse primers, and amplicon sizes used in this study are provided in Table 1.

### 2.4. Determination of Thioredoxin Reductase Activity (µmol/min/mL)

We homogenized approximately 0.2 g of tissue from *P. major* and liver in 1 mL cold 50 mM potassium phosphate buffer, containing 1mM Ethylenediaminetetraacetic acid (EDTA) at pH 7. Following centrifugation (10,000× *g* for 15 min) at 4 °C, the supernatants were stored frozen at −86 °C until analysis. We determined the activity of TXNRD (Cayman Chemical Company, Ann Arbor, MI, USA, Item No.10007892) spectrophotometrically at 405 nm using a Spectra Max 5 microplate reader by Molecular Devices (Sunnyvale, CA, USA).

### 2.5. Determination of Peroxiredoxin1 Levels (pg/mL) 

Roughly 0.1 g of tissue from the *P. major* and liver were homogenized in 1 mL phosphate buffer saline, pH 7 on ice and lysed by ultra-sonication. After centrifugation (5000× *g* for 5 min), the supernatants were collected for assaying Prdx1 (LifeSpan BioSciences, Washington DC, USA, Items No. LS-F19515) spectrophotometrically via SpectraMax5 microplate reader (Molecular Devices, Sunnyvale, CA, USA) at a wavelength of 450 nm.

### 2.6. Statistical Analysis

Data analyses for TXNRD, Prdx1 and DNA methylation were carried out using PROC GLM in SAS [21]. Contrasts between the treatment levels were used to assess the statistical significance.

## 3. Results

We investigated the effects of short and long term HS on the TXN–Prdx and the TXN systems in meat-type chickens. The expressions of *TXN* and *TXNRD* are presented in Figure 1.

The relative levels of mRNA for the *TXN* gene in liver tissues was down-regulated after 1 and 12 days of exposure to HS and this down-regulation was statistically significant (*p* < 0.05) at the 12 day time point only. Meanwhile, the *TXNRD* was slightly up-regulated only at 12 days post HS, but this was not statistically significant (*p* < 0.1). *Prdx1*, *Prdx3*, and *Prdx4* expressions are presented in Figure 2.

Compared to controls, mRNA levels of *Prdx1*, *Prdx3*, *and Prdx4* were down-regulated at day 1 (*p* < 0.1) and day 12 (*p* < 0.05) of exposure to HS, but this down-regulation was statistically significant only at day 12 post-HS.

The cellular activity level of TXNRD, and the cellular level of Prdx1 and DNA methylation are presented in Figure 3, Figure 4 and Figure 5.

The TXNRD activity levels decreased (*p* < 0.01) in the liver only at 12 days post-HS. There was no statistically significant difference in the TXNRD levels in the *P. major* both at 1 and 12 days post-HS (Figure 3). Additionally, heat stressed birds showed a decrease (*p* < 0.01) in the Prdx1 level at 12 days post-HS in the liver tissue and at 1 day post-HS in the *P. major* muscle (Figure 4). The concentration of methylated DNA was slightly but not significantly decreased after 1 day in *P. major* (*p* > 0.05) but it was significantly decreased in liver at 12 days in the stressed group (*p* < 0.05) (Figure 5). Meanwhile, after 1 day of HS, the quantity of DNA methylation in liver was higher in the heat stressed group than the control group, but not significant.

## 4. Discussion

HS is one of the most important burdens to the poultry industry, causing major economic losses. It is known that HS causes an adverse effect on the growth and production performance of poultry [1,22]. An understanding of the change in the expression of the TXN system and the cellular responses to HS could offer a new strategy to mitigate the effect of HS.

### 4.1. Heat Stress and Expression of the Thioredoxin System

The TXN system consisting of TXN, and TXNRD plays an important role in oxidative and nitrosative stress mitigation through reducing and scavenging peroxides and free radicals [23]. TXN reduces disulfide bonds resulting from protein oxidation in a reaction that results in oxidized TXN, which is then turned over by TXNRD using NADPH [24]. This system, in combination with Prdx activity, plays an important role in consuming mitochondrial H_2_O_2_ [25]. Data on the regulation of *TXN*, *TXNRD*, *Prdx1*, *Prdx3*, and *Prdx4* in both acute and chronic HS is scant. In the current study, the *TXN*, *Prdx1*, *Prdx3*, and *Prdx4* genes mRNA expressions were down-regulated at day 12 post-HS. Thus, chronic HS causes changes in the TXN-Prdx system that are not apparent in acute HS. This might be due to the increase in H_2_O_2_.

Treatment of SH-SY5Y human neuroblastoma cells with different concentrations of H_2_O_2_ led to decreased mRNA expression of *TXN*, which was exacerbated by induced mitochondrial dysfunction [26]. Thus, the increased cellular H_2_O_2_ as a result of HS may be responsible for the down-regulation of *TXN*. In the current work, *TXNRD* expression is somewhat increased in the heat stressed group at day 12 post-HS, perhaps as a compensatory mechanism in the face of increased oxidative stress. However, the significant down-regulation of *TXN* shown in the heat stressed group could limit the substrate available for reaction of TXNRD and thus limit its protective abilities.

Peroxiredoxins are ubiquitous cellular enzymes that scavenge peroxide and peroynitrite [27]. The mRNA expression of *Prdx1*, *Prdx3*, and *Prdx4* significantly decreased at 12 days post-HS. Kang et al. showed that there are six isoforms of Prdx [28]. Prdxs1–4 are TXN peroxidases that require TXN as an electron donor to remove H_2_O_2_ and contain two conserved cysteine residues that participate in the reaction [29]. It is apparent that exposure of chicks to HS leads to mRNA expression changes in *TXN* and elicits down-regulation of *Prdx*1, 3, and 4 at day 12 post-HS. Hwang et al. showed that *Prdx3* increased in neurons 1 day post ischemia and in glia 3 days post ischemia related to removal of ROS and phagocytic processes [30]. They also showed increases in TXN2 30 min post ischemia, which decreases 6 h later and increases again 1 day post ischemia indicating time sensitivity of TXN system changes. The decrease in TXN was correlated with cell death. Infusion with Prdx3 alone or in combination with TXN2 protects against ischemic damage and reduces lipid peroxidation. The results of Hwang et al. [30] and that of the current study may indicate acute compensation followed by long-term decompensation, leading to decreased antioxidant activity and may be one reason why HS causes decreases in performance. These issues may be remedied by antioxidant replacement and require further study.

### 4.2. Heat Stress and Cellular Activity of the Thioredoxin System

HS has been shown to elevate the level of ROS in several animal models, and the TXN-Prdx system is one of the main hydrogen peroxide-scavenging systems used by cells [13,14,31]. TXNRD enzymes catalyze the reduction of TXN to restore its antioxidant activity. In the current study, the liver TXNRD activity and Prdx1 levels decreased under chronic HS but not under acute stress. The TXNRD and Prdx1 response to HS may depend on the duration of the stress and also the tissue examined. Based on our results, it seems that liver tissue TXN-Prdx system is more affected by chronic HS than the muscle, and this may be due to the liver’s larger role in scavenging free radicals.

### 4.3. Heat Stress and DNA Methylation

This study also involved measurement of the levels of DNA methylation, because DNA methylation plays a key role in the regulation of gene expression, and chronic oxidative stress is known to change the expression of DNA methylation regulators [32,33]. The TXN system genes have increased DNA methylation levels in the liver of Wilson’s disease patients and in the disease mouse model where copper induced oxidative stress is a major factor [34]. In the current study, with an increase in the duration of exposure to HS, the concentrations of global DNA methylation were down-regulated significantly in the liver. Meanwhile, in *P. major*, the concentrations of DNA methylation were not significantly different after 1 or 12 days of HS than the control. HS-induced oxidative stress may result in global hypomethylation but hypermethylation of specific genes. The methylation status of antioxidant genes under HS needs further study. Mordaunt et al. [34] also showed that choline supplementation can increase TXN and Prdx gene expression in mouse liver, suggesting that choline supplementation could be pursued as a remedy for the decreased expression of these two genes we studied in the livers of chronically heat stressed birds.

## 5. Conclusions

There are several studies that have evaluated very short-term effects of HS. In the current study, we show that molecular responses of TXN-Prdx system to short term exposure to HS may be different from that of relatively long term. We show that exposure of broilers to HS for longer times leads to decreased gene expression of TXN, Prdx1, Prdx3, and Prdx4. Furthermore, HS decreases the level of TXNRD, Prdx1, and DNA methylation in liver. Taken together, we suggest that TXNRD, Prdx1 and DNA methylation can be used as biomarkers of oxidative damage in meat-type chickens under chronic HS. The implications of the dynamics of genes in the ROS and TXN-Prdx pathways may provide insight into nutritional intervention strategies (e.g., supplementation of antioxidants) when birds are exposed to HS.

## Figures and Tables

**Figure 1 animals-11-01957-f001:**
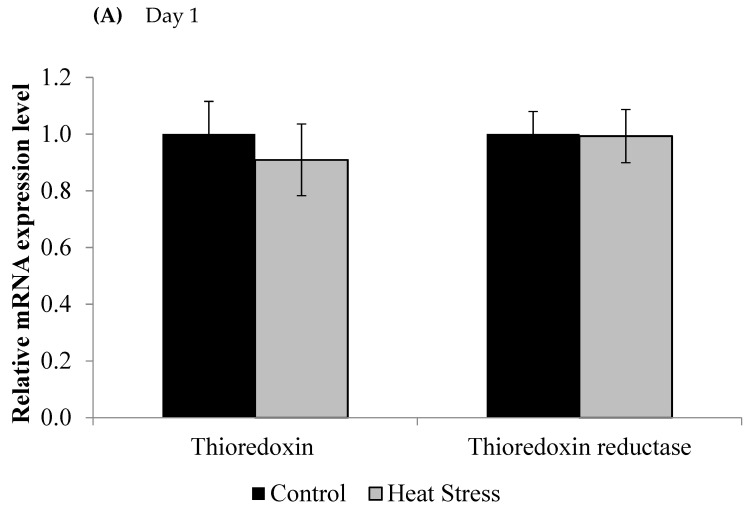
Effect of heat stress on thioredoxin and thioredoxin reductase mRNA at day 1 (**A**) and day 12 (**B**) in liver tissue of broiler. (** *p* < 0.01 and ^+^
*p* < 0.1).

**Figure 2 animals-11-01957-f002:**
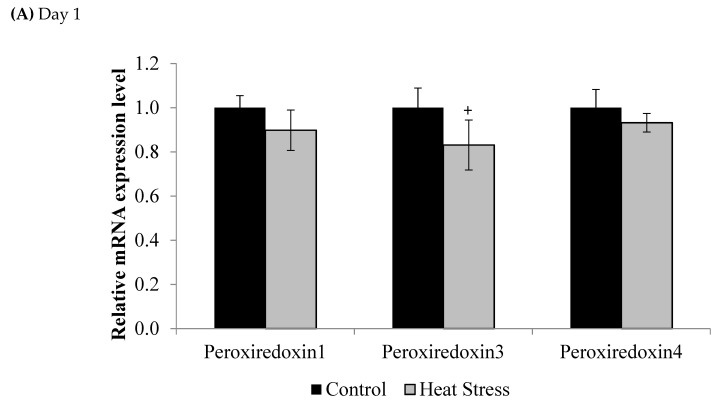
Effect of heat stress on *peroxiredoxin 1*, *peroxiredoxin 3*, and *peroxiredoxin 4* mRNA at day 1 (**A**) and day 12 (**B**) in liver tissue of broiler. (** *p* < 0.01 and ^+^
*p* < 0.1).

**Figure 3 animals-11-01957-f003:**
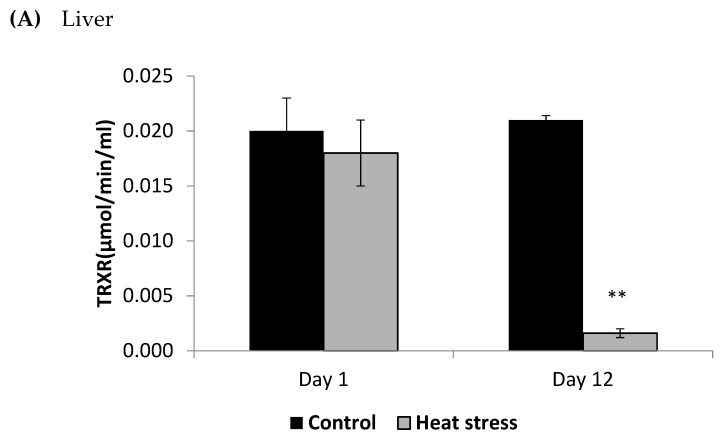
Effect of heat stress on thioredoxin reductase (TRXR) concentration in liver (**A**) and *Pectoralis major* (**B**) tissue of broiler (** *p* < 0.01).

**Figure 4 animals-11-01957-f004:**
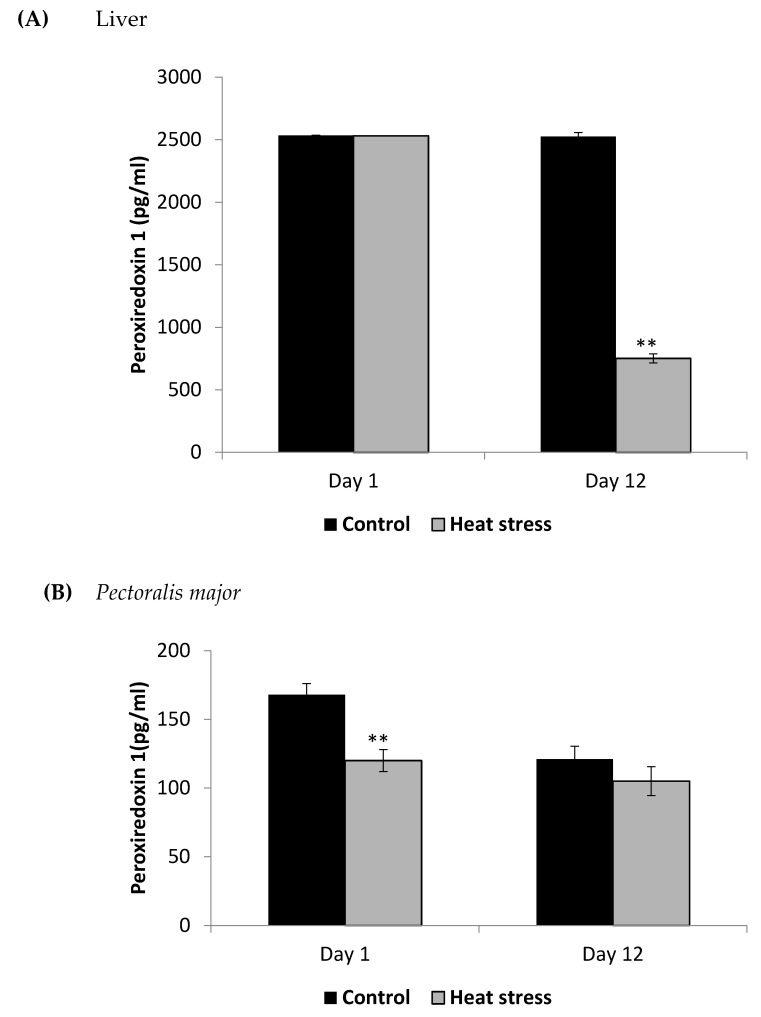
Effect of heat stress on peroxiredoxin1 concentration in liver (**A**) and *Pectoralis major* (**B**) tissue of broiler (** *p* < 0.01).

**Figure 5 animals-11-01957-f005:**
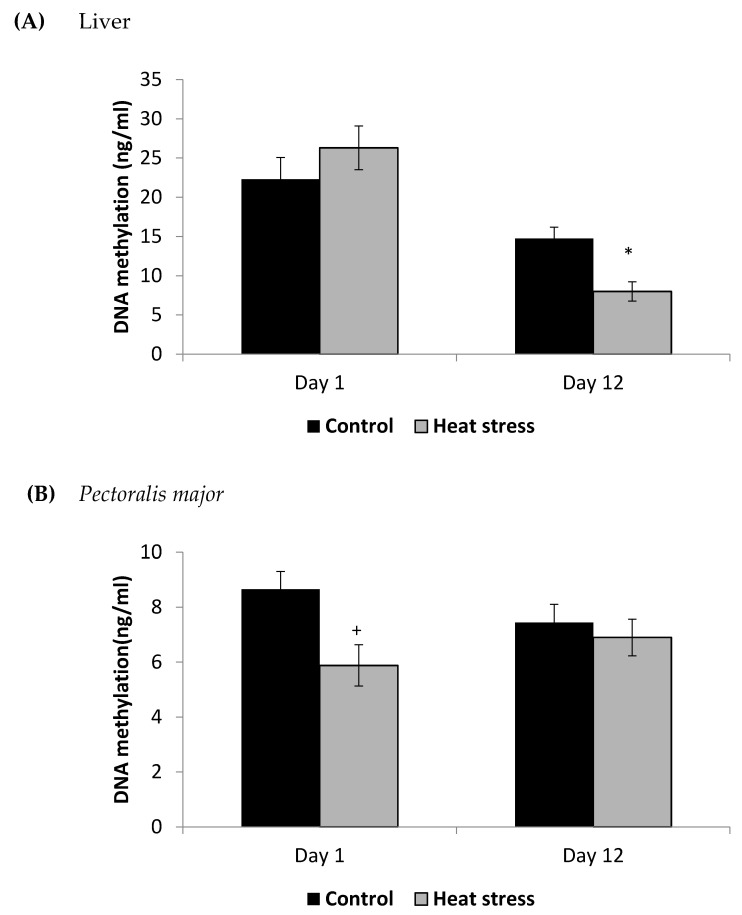
Effect of heat stress on DNA methylation in liver (**A**) and *Pectoralis major* (**B**) tissue of broiler (* *p* < 0.05 and ^+^
*p* < 0.1).

**Table 1 animals-11-01957-t001:** Primer pairs used to analyze gene expression by quantitative real time-PCR, and size of product.

Gene Name and Symbol	Gene Bank Accession Number	Product Size (bp)	Primer Sequence	
*Thioredoxin* *(TXN)*	NM_205453.1	82	Forward	5′CATGCCAACATTCCAGTTCTAC 3′
			Reverse	5′GGTCTCTTCCAGCTTCTCTTT 3′
*Thioredoxin reductase1* *(TXNRD1)*	NM_001030762.2	126	Forward	5′AGAGCATGACCCAGCTTTATT 3′
			Reverse	5′GTGTGAAGGAAGCCTCAGTATC 3′
*Peroxiredoxin 1* *(PRDX1)*	NM_001271932.1	84	Forward	5′GTACAGTGACAGAGCTGATGAA 3′
			Reverse	5′GCAAGGTGACAGAAGTGAGA3′
*Peroxiredoxin 3* *(PRDX3)*	XM_426543.5	95	Forward	5′GGAAATACCTCGTGCTCTTCTT 3′
			Reverse	5′GTGGAACTCATTCGCTTTGTTAC 3′
*Peroxiredoxin 4* *(PRDX4)*	XM_416800.5	100	Forward	5′GGACTCGGACCAATGAAGATT3′
			Reverse	5′CCCTAAGTGCATGTCCTTGAT 3′
*β-actin*	NM_205518.1	125	Forward	5′AGACATCAGGGTGTGATGGTTGGT3′
			Reverse	5′TCCCAGTTGGTGACAATACCGTGT3′

## Data Availability

The datasets generated during and/or analyzed during the current study are available from the corresponding author on reasonable request.

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
