# Peer review of "Molecular and Cellular Responses of DNA Methylation and Thioredoxin System to Heat Stress in Meat-Type Chickens"

_animals, 2021, doi:10.3390/ani11071957_

Round 1
Reviewer 1 Report
This is a well written manuscript describing the effect of short and long term exposure to heat stress on DNA methylation, gene expression and cellular activity of Thioredoxin and peroxiredoxin proteins system in chicken. The Authors analyzed the expression of proteins related to heat stress on Pectoralis major muscle and liver tissue of chicken strain Cobb500.
Here are some suggestion that the Authors could consider improving the article.
Introduction
Okay, but it is missing to talk about other studies on acute and chronic heat stress in chicken and on protein involved, which appears in the literature.
Line 78: specify if Cobb500 is a slow or fast growing broiler strand.
Line 79: please modify 25oC and 35oC in “25°C” and “35°C”.
Line 80: why the Authors apply the heat stress temperature for only 12 days? In most of paper the heat stress temperature was applied for 30 to 48 days in order to obtain a good chronic stress effect (see Rimoldi et al 2015 https://doi.org/10.1016/j.mgene.2015.08.003; Azad et al 2010 https://doi.org/10.1016/j.cbpa.2009.12.011)
Line 87: please modify -20oC in “-20°C”.
Line 237-244: I think that only 12 days of heat stress are not enough for stimulate a response of muscle tissue.
Please add to the reference Rimoldi et al., 2015 https://doi.org/10.1016/j.mgene.2015.08.003.
Author Response
We thank the reviewers for their time spent carefully reviewing the manuscript, and in their opinions. In what follows the reviewers’ comments are in black and the authors’ responses are in blue
Reviewer 1:
Okay, but it is missing to talk about other studies on acute and chronic heat stress in chicken and on protein involved, which appears in the literature.
Line 78: specify if Cobb500 is a slow or fast growing broiler strand.
Thank you for pointing this out. The reviewer is correct, and we have specified that cobb 500 is a fast-growing broiler in line 81
Line 79: please modify 25oC and 35oC in “25°C” and “35°C”.
As suggested by the reviewer, we have modify this in all the manuscript
Line 80: why the Authors apply the heat stress temperature for only 12 days? In most of paper the heat stress temperature was applied for 30 to 48 days in order to obtain a good chronic stress effect (see Rimoldi et al 2015 https://doi.org/10.1016/j.mgene.2015.08.003; Azad et al 2010 https://doi.org/10.1016/j.cbpa.2009.12.011)
Thank you for this suggestion. It would have been interesting to explore this aspect. However, as the definition of chronic heat stress is exposure to heat stress for several days. There are a lot of paper showed that exposure of chickens to HS for 7 or 14 days markedly affected on performance ( See Garriga ,et al., 2006 https://journals.physiology.org/doi/full/10.1152/ajpregu.00393.2005; Jastrebski et al., 2017https://journals.plos.org/plosone/article?id=10.1371/journal.pone.0181900)
Line 87: please modify -20oC in “-20°C”.
As suggested by the reviewer, we have modify this in line 87
Line 237-244: I think that only 12 days of heat stress are not enough for stimulate a response of muscle tissue.
Thank you for this suggestion. It would have been interesting to explore this aspect. However, in previous study we found that muscle tissue exhibited higher levels of protein oxidation when the birds exposed to acute and chronic heat stress. Also, Ma et al., 2021 reported that exposure to chronic HS for 1 wk elevated muscle atrophy(https://www.ncbi.nlm.nih.gov/pmc/articles/PMC7772709/).
Please add to the reference Rimoldi et al., 2015 https://doi.org/10.1016/j.mgene.2015.08.003.
Thank you for pointing this out. We have put this ref. no 22 line 212

Reviewer 2 Report
In their manuscript, Habashy et al. present a study in the effects of heat stress in broilers on the thioredoxin system and DNA-methyation. The study is well dsigned and conducted and the manuscript is well written. In the end of the conclusion, the authors allude to possible implications of their results for nutritional intervention strategies . This comes short in the discussion. Maybe the authors could be a bit clearer on what they mean here.
I have some minor formal remarks:
- Please take care of differentiating between proteins and genes and put the latter in italics.
- Please check for super-/subscripts to be written correctly (e.g. in H2O2)
- The heading in line 245 should be bold faced, I guess
Author Response
We thank the reviewers for their time spent carefully reviewing the manuscript, and in their opinions. In what follows the reviewers’ comments are in black and the authors’ responses are in blue
Reviewer 2:
In their manuscript, Habashy et al. present a study in the effects of heat stress in broilers on the thioredoxin system and DNA-methyation. The study is well dsigned and conducted and the manuscript is well written. In the end of the conclusion, the authors allude to possible implications of their results for nutritional intervention strategies . This comes short in the discussion. Maybe the authors could be a bit clearer on what they mean here.
I have some minor formal remarks:
- Please take care of differentiating between proteins and genes and put the latter in italics.
Thank you for pointing this out. We have different this in all the manuscript
- Please check for super-/subscripts to be written correctly (e.g. in H2O2)
As suggested by the reviewer, we have modify this in all the manuscript
- The heading in line 245 should be bold faced, I guess
We think this is an excellent suggestion. We have made it bold.
